# Automatic Synthesis of High-quality Triplet Data for Composed Image Retrieval

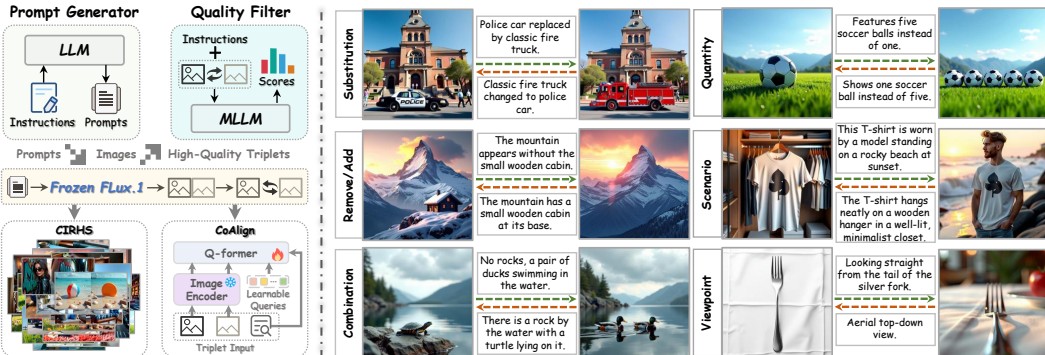

**a) High-quality Triplets Synthesis**    **b) Representative examples in the CIRHS dataset**

Figure 1: **Overview of our key innovations.** (a) Illustration of the proposed automatic triplet synthesis pipeline, along with our proposed training framework CoAlign. (b) Representative samples from our CIRHS dataset, covering various real-world scenes and objects, as well as diverse editing operations including object or scene change, quantity variation, and viewpoint shift, etc.

## Abstract

As a challenging vision-language (VL) task, Composed Image Retrieval (CIR) aims to retrieve target images using multimodal (image+text) queries. Although many existing CIR methods have attained promising performance, their reliance on costly, manually labeled triplets hinders scalability and zero-shot capability. To address this, we propose a scalable pipeline for automatic triplet generation, along with a fully synthetic dataset named Composed Image Retrieval on High-quality Synthetic triplets (CIRHS). Our pipeline leverages a large language model (LLM) to generate diverse prompts, controlling a text-to-image generative model to produce image pairs with identical elements in each pair, which are then filtered and reorganized to form the CIRHS dataset. In addition, we introduce Hybrid Contextual Alignment (CoAlign), a novel CIR framework, which can accomplish global alignment and local reasoning within a broad context, enabling the model to learn robust and informative representations. By utilizing the synthetic CIRHS dataset, CoAlign achieves outstanding zero-shot performance on three commonly used benchmarks, demonstrating for the first time the feasibility of training CIR models on a fully synthetic dataset. Furthermore, under supervised training, our method outperforms the state-of-the-art supervised CIR approaches, validating the effectiveness of our proposed retrieval framework. The code and the CIRHS dataset will be open-sourced.

## 1 Introduction

Composed Image Retrieval (CIR) (Vo et al., 2019; Wu et al., 2021; Liu et al., 2021) has attracted increasing attention in recent years, aiming to retrieve target images based on a query consisting of a reference image and a relative caption. By integrating information from both modalities, CIR can attain more precise and flexible searches, and provide a superior user experience compared

with conventional unimodal retrieval (Datta et al., 2008). With the emergence of large-scale vision-language pretraining models (VLMs) (Radford et al., 2021; Alayrac et al., 2022; Li et al., 2023), CIR has made significant progress and found numerous applications.

Existing supervised CIR approaches (Baldrati et al., 2022; Liu et al., 2024; Xu et al., 2024) heavily rely on manually annotated triplets, which are both time-consuming and labor-intensive to construct. Moreover, due to the limited scale of available datasets such as FashionIQ (Wu et al., 2021) with only 46.6k triplets and CIRR (Liu et al., 2021) with just 28.8k, these methods suffer from poor generalization performance. Consequently, several studies present Zero-Shot Composed Image Retrieval (ZS-CIR). Early approaches (Saito et al., 2023; Baldrati et al., 2023) employ an inversion network (Gal et al.) trained on massive image-text pairs. However, these methods have the inherent task discrepancy (Byun et al., 2024), making them suboptimal solutions. Additionally, some training-free approaches (Karthik et al., 2024; Yang et al., 2024b) introduce large language model (LLM) reasoning into ZS-CIR. While promising, they often fail to capture fine-grained visual details and the high complexity of model architecture makes it infeasible to conduct domain-specific fine-tuning, thus limiting their applicability. Recent studies (Ventura et al., 2024; Gu et al., 2024a) have designed automated pipelines for creating large-scale triplet datasets, and they also unify the model architecture for both ZS-CIR and supervised CIR. Compared with previous methods, this line of work enables more robust generalization while preserving the ability to fine-tune on domain-specific datasets. However, they face two limitations: (1) The relative captions cover only a narrow range, primarily focusing on substitution, which results in a lack of diversity. (2) The reference and target images produced by image editing (Brooks et al., 2023) are often of low quality, with severe artifacts and unrealistic appearances.

To deal with the above limitations, we propose a scalable pipeline for automatic triplet synthesis as illustrated in Figure 1(a), which generates high-quality CIR triplets in three stages. In Stage 1, an LLM is employed to generate numerous textual quadruplets, each consisting of two image captions as well as two relative captions that describe how one image can be transformed into the other. Guided by a carefully crafted instruction with randomized elements, the LLM-generated image captions cover a wide range of real-world objects and scenes, while the relative captions include diverse editing operations such as substitution, removal, and composition, thus effectively relieving the first issue mentioned before. However, textual quadruples are insufficient for training CIR models, as the LLM-generated image captions need to be converted into corresponding images.

Therefore, in Stage 2, we focus on how to obtain high-quality images corresponding to the textual quadruples. A straightforward solution is to use a text-to-image generative model (T2I-GM) to independently synthesize an image from each caption. However, this will inevitably faces a drawback: CIR requires consistency in shared elements between the reference and target images, while independent generation may result in uncontrollable visual discrepancies. We notice that T2I-GMs are capable of preserving strong intra-image consistency. Consequently, we combine the two image captions in each textual quadruple into a single prompt using a predefined template. This prompt is fed into the T2I-GM to generate an image containing two semantically related sub-images in a single forward pass, which are cropped to serve as the reference and target images, respectively.

Stage 3 performs data filtering, where the generated images and relative captions are reorganized into CIR triplets. These triplets are scored by a multimodal large language model (MLLM) based on three criteria: image-text fidelity, image quality and triplet alignment. Afterwards, we compute a weighted sum of the three scores and discard the bottom 15% of the triplets, resulting in a high-quality and fully synthetic dataset of 534k triplets, namely Composed Image Retrieval on High-quality Synthetic triplets (CIRHS), with representative samples shown in Figure 1(b).

Moreover, we propose a unified framework, Hybrid Contextual Alignment (CoAlign), for both supervised and zero-shot CIR. CoAlign optimizes the model within a broad context, combining global and local objectives to learn robust and fine-grained representations. Extensive experiments on three popular benchmarks validate the effectiveness of our method under both supervised and zero-shot CIR settings, as well as the feasibility of training CIR models using purely synthetic data. To sum up, our contributions are threefold:

• We propose a scalable pipeline for automatic CIR triplet synthesis, tackling previous limitations such as low image quality, unrealistic appearances, and the lack of diversity in relative captions. With

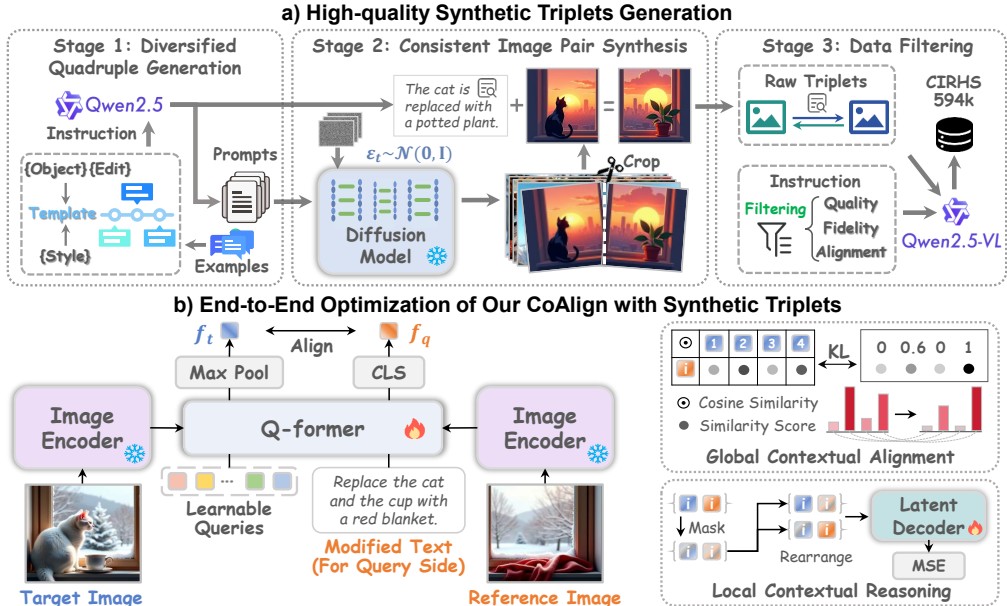

Figure 2: **Overall framework of our method.** (a) The triplet synthesis pipeline involves three stages: generating diverse textual quadruples via an LLM, synthesizing and reorganizing consistent image pairs into triplets, and filtering low-quality samples using an MLLM. (b) The model architecture of CoAlign. The left side illustrates the encoding process of the query and target using different modes, while the right shows the global and local optimization objectives employed by CoAlign.

this pipeline, we obtain a large-scale, fully synthetic CIR dataset named CIRHS, which consists of 534k high-quality triplets.

• We propose a novel CIR framework, Hybrid Contextual Alignment (CoAlign), which optimizes the model within a broad context by combining global alignment and local reasoning. It is simple yet effective and can enhance the robustness of learned representations.

• Experiments show the superior performance under both supervised and zero-shot CIR settings. To the best of our knowledge, this is also the first work to verify the feasibility of training CIR models entirely on synthetic data.

## 2 RELATED WORK

**Composed Image Retrieval (CIR)** is primarily evaluated on the fashion domain (Wu et al., 2021) and real-world scenarios (Liu et al., 2021; Baldrati et al., 2023). Mainstream methods (Baldrati et al., 2022; Liu et al., 2024; Xu et al., 2024) leverage the cross-modal alignment of VLMs and apply early or late fusion to integrate the two modalities in the composed query. Recently, zero-shot CIR has gained attention, with textual inversion (Saito et al., 2023; Baldrati et al., 2023; Gu et al., 2024b; Gal et al.) becoming a key technique. This method maps the input image to a pseudo-word token, which is then combined with the relative caption and encoded by a text encoder. Other works (Karthik et al., 2024; Yang et al., 2024b) use LLMs for target caption generation, reformulating CIR as text-to-image retrieval. However, their complex architectures hinder domain-specific fine-tuning and practical deployment. Due to the lack of labeled triplets, recent efforts have focused on automatically constructing CIR triplets. Some methods (Ventura et al., 2024; Levy et al., 2024) derive similar image pairs from public databases and generate relative captions via handcrafted rules or LLMs. CompoDiff (Gu et al., 2024a) and VISTA (Zhou et al., 2024) synthesize target images through editing (Brooks et al., 2023), but they are limited by the quality of the generated images. In contrast, our method generates high-quality, diverse triplets with photorealistic reference and target images.

**Text-to-Image Generation** has evolved from early GAN-based methods (Goodfellow et al., 2020) to more complex multimodal frameworks. Currently, diffusion models (Ho et al., 2020) dominate

tasks like text-to-image synthesis (Podell et al., 2023b; Ramesh et al., 2022b), image translation (Saharia et al., 2022), and controllable generation (Zhang et al., 2023; Mou et al., 2024). Notably, latent diffusion models (LDM) (Rombach et al., 2022) improve image-text fidelity while reducing computational cost, paving the way for excellent works supporting high-resolution image generation, such as Stable Diffusion (Podell et al., 2023a) and DALL-E (Ramesh et al., 2022a). Recently, diffusion transformers (DiT) (Peebles & Xie, 2023) further enhance scalability, with advanced models like PixArt (Chen et al., 2024a) and Flux (Labs, 2024) achieving state-of-the-art generation quality.

## 3 METHODOLOGY

As shown in Figure 2, our method consists of two parts: 3.1 presents our automatic triplet synthesis pipeline, including textual quadruple generation, consistent image pair synthesis, and data filtering, while 3.2 describes our proposed CIR framework, CoAlign, detailing its model architecture, optimization strategy, and inference workflow.

### 3.1 HIGH-QUALITY TRIPLET SYNTHESIS FOR CIR

**Diverse Quadruple Generation.** End-to-end generation of CIR triplets is highly challenging. Therefore, in Stage 1, we first generate their textual counterparts. Specifically, we design an instruction template $\mathcal{P}(object, edit, style)$ to guide an LLM $g_{llm}(\cdot)$ in producing textual quadruples, as shown below. The three parameters are sampled from predefined sets crafted by GPT-4o (OpenAI et al., 2024), covering diverse objects, editing operations, and image styles.

$$g_{llm}(p) \to \langle C_{I_r}, C_{r \to t}, C_{t \to r}, C_{I_t} \rangle, \tag{1}$$

where the reference caption $C_{I_r}$ and the target caption $C_{I_t}$ are used to synthesize $I_r$ and $I_t$, sharing at least one semantic entity, while the relative caption $C_{r \to t}$ captures the modification from $I_r$ to $I_t$. Additionally, the inverse caption $C_{t \to r}$ describes the change from $I_t$ to $I_r$, enabling bidirectional triplet construction and thereby improving efficiency. The instruction $p \sim \mathcal{P}$ varies with each input, encompassing various common objects and editing operations such as object composition and scenario change. Furthermore, style information is embedded in $C_{I_r}$ and $C_{I_t}$ for generation across multiple domains. Below is an illustration of the template $\mathcal{P}$.

> Using the elements: {**suggested objects**}. {**editing operations**}. {**image styles**}, please help me generate a quadruple that meets the requirements of CIR.

**Consistent Image Pair Synthesis.** After obtaining the textual quadruplets, a T2I-GM can synthesize $I_r$ and $I_t$ using $C_{I_r}$ and $C_{I_t}$, respectively. However, it doesn't ensure consistency of shared elements between the two generated images. That is, $I_r$ and $I_t$ may differ significantly, making them unsuitable for constructing CIR triplets.

> HD 4k square grid layout for left and right images, Left: {$C_{I_r}$}, Right: {$C_{I_t}$}.

In contrast, we leverage the inherent consistency of generative models, i.e., ***they have the ability to generate identical elements within a single image.*** To this end, we define a prefix to specify the desired image layout and integrate $C_{I_r}$ and $C_{I_t}$ into a single prompt as shown above, which is then fed into the T2I-GM to generate a single image containing two side-by-side sub-images. $I_r$ and $I_t$ are finally obtained by cropping the left and right parts, as illustrated in Figure 2(a).

To fully leverage the textual quadruples, we synthesize $n$[1] image pairs for each $(C_{I_r}, C_{I_t})$ based on different random seeds, yielding $\{(I_r^i, I_t^i)\}_{i=1}^n$. By combining these images with their relative captions, we obtain $2n$ CIR triplets, i.e., $\{(I_r^i, C_{r \to t}, I_t^i)\}_{i=1}^n \cup \{(I_t^i, C_{t \to r}, I_r^i)\}_{i=1}^n$. Additionally, we introduce an identifier, namely triplet identity (TID), and assign the same TID to triplets sharing the same relative caption. Triplets with the same TID exhibit a certain degree of similarity, making label smoothing (Müller et al., 2019) possible during training.

**Data Filtering.** To refine the raw synthesized triplets, Stage 3 incorporates a multimodal large language model (MLLM) (Bai et al., 2025) to score them across three aspects on a scale from 1 to

---

[1]Considering scalability and efficiency, we select $n = 10$.

Table 1: **Performance comparison with existing supervised CIR methods.** The best results are marked in bold, and the second-best results are underlined. † indicates that the method is pretrained on its own constructed triplet dataset.

| Method | CIRR | | | | FashionIQ | | | | | | | |
| | Recall@K | | Recall$_s$@K | | Dress | | Shirt | | Toptee | | Average | |
| | K=1 | K=5 | K=1 | K=3 | K=10 | K=50 | K=10 | K=50 | K=10 | K=50 | K=10 | K=50 |
|---|---|---|---|---|---|---|---|---|---|---|---|---|
| TIRG (Vo et al., 2019) | 14.61 | 48.37 | 22.67 | 65.14 | 14.87 | 34.66 | 18.26 | 37.89 | 19.08 | 39.62 | 17.40 | 37.39 |
| MAAF (Dodds et al., 2020) | 10.31 | 33.03 | 21.05 | 61.60 | 23.80 | 48.60 | 21.30 | 44.20 | 27.90 | 53.60 | 24.30 | 48.80 |
| CIRPLANT (Liu et al., 2021) | 19.55 | 52.55 | 39.20 | 79.49 | 17.45 | 40.41 | 17.53 | 38.81 | 61.64 | 45.38 | 18.87 | 41.53 |
| ARTEMIS (Delmas et al., 2024) | 16.96 | 46.10 | 39.99 | 75.67 | 27.16 | 52.40 | 21.78 | 43.64 | 29.20 | 53.83 | 26.05 | 50.29 |
| CLIP4CIR (Baldrati et al., 2022) | 38.53 | 69.98 | 68.19 | 94.17 | 33.81 | 59.40 | 39.99 | 60.45 | 41.41 | 65.37 | 38.32 | 61.74 |
| TG-CIR (Wen et al., 2023) | 45.25 | 78.29 | 72.84 | 95.13 | 45.22 | 69.66 | 52.60 | 72.52 | 56.14 | 77.10 | 51.32 | 73.09 |
| Re-ranking (Liu et al., 2023) | 50.55 | 81.75 | 80.04 | 96.58 | 48.14 | 71.43 | 50.15 | 71.25 | 55.23 | 76.80 | 51.17 | 73.13 |
| BLIP4CIR+Bi (Liu et al., 2024) | 40.15 | 73.08 | 72.10 | 95.93 | 42.09 | 67.33 | 41.76 | 64.28 | 46.61 | 70.32 | 43.49 | 67.31 |
| CASE† (Levy et al., 2024) | 48.68 | 79.98 | 76.39 | 95.86 | 47.44 | 69.36 | 48.48 | 70.23 | 50.18 | 72.24 | 48.79 | 70.68 |
| CoVR† (Ventura et al., 2024) | 49.69 | 78.60 | 75.01 | 93.16 | 44.55 | 69.03 | 48.43 | 67.42 | 52.60 | 74.31 | 48.53 | 70.25 |
| CompoDiff† (Gu et al., 2024a) | 32.39 | 57.61 | 67.88 | 94.07 | 38.39 | 51.03 | 41.68 | 56.02 | 45.70 | 57.32 | 39.81 | 51.90 |
| CaLa (Jiang et al., 2024) | 49.11 | 81.21 | 76.27 | 96.46 | 42.38 | 66.08 | 46.76 | 68.16 | 50.93 | 73.42 | 46.69 | 69.22 |
| SPRC (Xu et al., 2024) | 51.96 | 82.12 | 80.65 | 96.60 | 49.18 | **72.43** | 55.64 | 73.89 | 59.35 | 78.58 | 54.72 | 74.97 |
| **CoAlign (Ours)** | **54.07** | **83.81** | **80.87** | **97.04** | **49.43** | 72.04 | **56.48** | **75.61** | 58.85 | **78.99** | **54.92** | **75.55** |

10: (1) image quality of both $I_r$ and $I_t$, including the clarity, noise, artifacts, etc. (2) image-text fidelity (e.g., $I_r \leftrightarrow C_{I_r}$), and (3) triplet alignment (e.g., $I_r + C_{r \to t} \leftrightarrow I_t$). The average score is computed via a weighted sum and a threshold $\alpha$ is then applied to filter out low-quality triplets, which accounts for about 15%.

Utilizing this pipeline, we build the large-scale CIRHS dataset with 534k high-quality synthetic triplets. Experiments will verify the scalability and effectiveness of this pipeline. Additional details on triplet synthesis, including the full prompts and more examples, are in Appendix B.

## 3.2 END-TO-END OPTIMIZATION WITH SYNTHETIC TRIPLETS

**CoAlign Model Architecture.** As shown in Figure 2(b), inspired by BLIP-2 (Li et al., 2023), our model consists of a frozen image encoder and a lightweight Querying Transformer (Q-Former), which incorporates learnable queries for efficient multimodal feature extraction. CoAlign reuses the Q-Former's two distinct encoding modes: image-grounded encoding (jointly conditioned on visual and textual inputs) and pure image encoding (visual-only processing).

Given an input triplet $\langle I_r, C_{r \to t}, I_t \rangle$, the query side uses the frozen image encoder to extract features from the reference image $I_r$. The resulting visual features, together with the relative caption $C_{r \to t}$, are forwarded to the Q-Former. The output [CLS] token is then passed through a query projection layer to produce the query feature $f_q \in \mathbb{R}^d$. Similarly, on the target side, the frozen image encoder processes the target image $I_t$, generating visual features that are then passed through the Q-Former in its pure image encoding mode. The output token embeddings corresponding to the learnable queries of the Q-Former undergo max-pooling across the sequence dimension, followed by a target projection layer, to obtain the target feature $f_t \in \mathbb{R}^d$.

**Hybrid Contextual Alignment.** To achieve comprehensive alignment between the composed query and its target images, CoAlign jointly conducts optimization from both global and local perspectives.

*Global Contextual Alignment.* Conventional contrastive learning (He et al., 2020; Oord et al., 2018) focuses on the diagonal elements of the similarity matrix. However, this becomes suboptimal for the proposed CIRHS dataset, where each query may correspond to multiple target images (same TID). To this end, CoAlign combines distribution matching (Zhang & Lu, 2018; Jiang & Ye, 2023) and label smoothing (Müller et al., 2019) to perform global contextual alignment, enabling the model to extract useful information within a broader context, i.e. the entire similarity matrix, rather than relying solely on the diagonal elements, thereby facilitating the learning of more robust representations. Specifically, for a mini-batch of size $N$, each query is associated with a set $\mathcal{S} = \{(f_q^i, f_t^j), y_{i,j}\}_{j=1}^N$, where $y_{i,j} = 1$ denotes a hard-matched pair, $y_{i,j} = \beta$, $\beta \in (0, 1)$ represents a soft-matched pair (sharing the same TID), and $y_{i,j} = 0$ indicates an unmatched pair. Then the matching probabilities

are computed via a softmax over $\mathcal{S}$:

$$p_{i,j} = \frac{\exp(sim(f_q^i, f_t^j)/\tau)}{\sum_{k=1}^N \exp(sim(f_q^i, f_t^k)/\tau)}, \tag{2}$$

where $sim(\cdot, \cdot)$ is the cosine similarity, and $\tau$ is a learnable temperature parameter that controls the sharpness of the probability distribution. The label distribution, representing the true matching probability, is computed as $q_{i,j} = y_{i,j}/\Sigma_{k=1}^N y_{i,k}$ and the global contextual alignment loss from query to target is calculated by the KL divergence:

$$\mathcal{L}_{q2t} = \frac{1}{N} \sum_{i=1}^N \sum_{j=1}^N p_{i,j} \log(\frac{p_{i,j}}{q_{i,j} + \epsilon}), \quad \mathcal{L}_{gca} = \mathcal{L}_{q2t} + \mathcal{L}_{t2q}, \tag{3}$$

where $\epsilon$ is used to prevent numerical issues. In the same way, $\mathcal{L}_{t2q}$ can be obtained by exchanging $f_q$ and $f_t$ in Equation 2, and the global contextual alignment loss $\mathcal{L}_{gca}$ is the bidirectional sum.

*Local Contextual Reasoning.* Complementary to global contextual alignment, we propose local contextual reasoning to capture finer-grained information within each triplet. Unlike masked language/image modeling (MLM/MIM) (He et al., 2022; Devlin et al., 2019), CoAlign adopts a lightweight decoder and performs bidirectional masked feature prediction (MFP) (Wei et al., 2022) at the latent level. For a composed query and its hard-matched target $(f_q, f_t)$, we first randomly mask out elements along the feature dimension with a probability of 30%. Following BERT (Devlin et al., 2019), the masked elements are replaced with 10% random, 10% unchanged, and 80% set to zero, yielding the masked pair $(\tilde{f}_q, \tilde{f}_t)$. Subsequently, a rearrange operation is performed to group and concatenate the features to obtain $[f_q, \tilde{f}_t] \in \mathbb{R}^{2d}$ and $[f_t, \tilde{f}_q] \in \mathbb{R}^{2d}$, which are passed through the latent decoder $\Phi$ (a two-layer MLP) to predict the masked elements. The reconstruction is supervised by the bidirectional local contextual reasoning loss $\mathcal{L}_{lcr}$:

$$\mathcal{L}_{lcr} = \mathbb{E}[||f_q - \Phi([f_t, \tilde{f}_q])||_2^2 + ||f_t - \Phi([f_q, \tilde{f}_t])||_2^2]. \tag{4}$$

The overall training objective $\mathcal{L}$ is a weighted sum of the global and local terms, where $\gamma$ is a hyperparameter,

$$\mathcal{L} = \mathcal{L}_{gca} + \gamma \mathcal{L}_{lcr}. \tag{5}$$

**Inference Workflow.** Given an image gallery with pre-extracted features $\mathcal{V} = \{f_t^j\}_{j=1}^N$, we compute the cosine similarity between a query $f_q$ and each $f_t^j$, returning the top-K most similar images as the retrieval results.

## 4 EXPERIMENTS

### 4.1 THE CIRHS DATASET

We construct CIRHS, a fully synthetic dataset containing 534k triplets. Table 2 summarizes its statistics. Compared with existing manually annotated datasets (Wu et al., 2021; Liu et al., 2021), CIRHS is significantly larger in scale. Although smaller than WebVid-CoVR (Ventura et al., 2024) and ST18M (Gu et al., 2024a), CIRHS offers advantages in quality and diversity. WebVid-CoVR is constrained by the lack of diversity in relative captions (mainly object or scene change). ST18M, on the other hand, based on image editing to generate CIR triplets, suffers from poor generation quality due to unrealistic outputs and visual artifacts. In contrast, CIRHS

Table 2: **Statistics of common CIR datasets.** We compare our CIRHS dataset with existing benchmarks.

| Dataset | Domain | Triplets | Images | Text length |
|---|---|---|---|---|
| CIRR | Natural | 36,554 | 21,185 | 59.51 |
| FashionIQ | Fashion | 30,132 | 7,988 | 27.13 |
| LaSCo | Natural | 389,305 | 121,479 | 30.70 |
| WebVid-CoVR | Natural | 1,644,276 | 130,559 | 23.36 |
| ST18M | Synthetic | 18,000,000 | - | - |
| **CIRHS (Ours)** | Synthetic | 534,758 | 534,758 | 53.17 |

Table 3: **Performance comparison with existing zero-shot CIR methods.** The best results are marked in bold, and the second-best results are underlined. † indicates that the dataset is synthetic.

| Method | Training Data | FashionIQ | | CIRR | | | | CIRCO | |
|---|---|---|---|---|---|---|---|---|---|
| | | R@10 | R@50 | R@1 | R@5 | $R_s$@1 | Avg. | mAP@5 | mAP@10 |
| PALAVRA (Cohen et al., 2022) | - | 19.76 | 37.25 | 16.62 | 43.49 | 41.61 | 42.55 | 4.61 | 5.32 |
| Pic2Word (Saito et al., 2023) | CC3M | 24.70 | 43.70 | 23.90 | 51.70 | - | - | - | - |
| SEARLE (Baldrati et al., 2023) | ImageNet1K | 27.61 | 47.90 | 24.87 | 52.31 | 53.80 | 53.06 | 11.68 | 12.73 |
| ContextI2W (Tang et al., 2024) | CC3M | 27.80 | 48.90 | 25.60 | 55.10 | - | - | - | - |
| KEDs (Suo et al., 2024) | CC3M | 26.80 | 47.90 | 26.40 | 54.80 | - | - | - | - |
| Slerp+TAT (Jang et al., 2024) | CC3M | 32.77 | 53.32 | 33.98 | 61.74 | 68.55 | 54.76 | 18.46 | 19.41 |
| Image2Sentence (Du et al.) | CC3M | 29.79 | 49.19 | 30.84 | 61.06 | - | - | 11.33 | 12.25 |
| CIReVL (Karthik et al., 2024) | - | 32.19 | 52.36 | 34.65 | 64.29 | 67.95 | 66.12 | **26.77** | **27.59** |
| *Comparison with methods based on CIR triplet construction* | | | | | | | | | |
| CoVR (Ventura et al., 2024) | WebVid-CoVR | 27.70 | 44.63 | 38.48 | 66.70 | 69.28 | 67.99 | 21.43 | 22.33 |
| CASE (Levy et al., 2024) | LaSCo+CoCo | - | - | 35.40 | 65.78 | 64.29 | 65.04 | - | - |
| CompoDiff (Gu et al., 2024a) | ST18M† | 39.02 | 51.71 | 26.71 | 55.14 | 64.54 | 59.84 | 15.33 | 17.71 |
| CLIP4CIR (Baldrati et al., 2022) | CIRHS† (Ours) | 26.94 | 47.73 | 29.64 | 62.16 | 57.78 | 59.97 | 20.17 | 21.98 |
| BLIP4CIR (Liu et al., 2024) | CIRHS† (Ours) | 30.89 | 52.74 | 25.76 | 55.12 | 55.08 | 55.10 | 18.73 | 20.02 |
| SPRC (Xu et al., 2024) | CIRHS† (Ours) | 37.44 | 57.91 | 38.32 | 68.93 | 69.34 | 69.14 | 21.76 | 23.12 |
| **CoAlign (Ours)** | CIRHS† (Ours) | **39.11** | **60.29** | **41.17** | **71.68** | **70.65** | **71.17** | 23.47 | 25.29 |

is designed to ensure both semantic diversity and high visual quality. Experiments also show that 534k triplets are sufficient to train strong CIR models. Note that the most time-consuming part, image generation, is a one-off process, taking an average of 7.2 seconds per image pair (i.e., two triplets) on a single H800 GPU. We will make all our data and code publicly available, so repeated consumption of computational resources will not be necessary.

## 4.2 EXPERIMENTAL SETUP

**Evaluation Benchmarks.** FashionIQ (Wu et al., 2021) simulates online shopping environment, with 30,134 triplets derived from 77,684 fashion-related images. CIRR (Liu et al., 2021) is the first open-domain dataset, containing 21,552 real-life images. CIRCO (Baldrati et al., 2023) builds on the COCO 2017 unlabeled split (Lin et al., 2014), with each query corresponding to multiple target images. More details about these datasets can be found in Appendix A.

**Comparison with ZS-CIR methods.** Table 3 compares existing zero-shot CIR methods. Our approach is the only one trained solely on synthetic triplets while achieving strong performance. Among methods based on CIR triplet construction, it outperforms all others across all metrics. Notably, CIRHS is compatible with any CIR framework, e.g., SPRC and CLIP4CIR also perform well when trained on it. On CIRCO, our method ranks second. This is primarily due to the large visual discrepancies between reference and target images inherent in CIRCO, where retrieval relies heavily on the relative caption. This reliance deviates from the original intent of CIR and makes it more favorable to training-free methods such as CIReVL. However, the complex architectures of such methods hinder domain-specific fine-tuning, whereas our approach supports it.

**Evaluation Metrics.** Recall@K is the main metric for CIRR and FashionIQ, with CIRR also reporting $Recall_s$@K on visually similar subsets and overall performance as $Avg. = \frac{Recall@5 + Recall_s@1}{2}$. For CIRCO, where each query has multiple targets, mAP@K is used as the primary metric.

**Implementation Details.** (1) We construct CIRHS using 8 H800 GPUs, with Qwen2.5-32B (Yang et al., 2024a) as the LLM, Flux.1-dev (Labs, 2024) as the T2I-GM, and Qwen2.5-VL-32B (Bai et al., 2025) as the MLLM for filtering. Each textual quadruple generates 10 side-by-side images at $512 \times 1056$ resolution, which are then cropped into $512 \times 512$ reference-target image pairs. The MLLM scores each triplet (scale 1-10) on three aspects, and a weighted sum (0.3, 0.2, 0.5) with threshold $\alpha = 7.5$ filters out 15% of low-quality triplets. (2) For CoAlign, we adopt BLIP-2 (Li et al., 2023) with a frozen ViT-G/14 (Dosovitskiy et al., 2020) (224px input), trained for 10 epochs on CIRHS using AdamW (LR 5e-6, batch size 128, $\beta = 0.6$, $\gamma = 0.4$) on a H800 GPU. For CIRR and FashionIQ, we train from scratch for 50 and 30 epochs with initial LRs of 1e-5 and 2e-5, respectively, using the same batch size 128.

Table 4: **Ablation experiments on CoAlign.** Best results are in bold. To validate GCA, we introduce the image-text contrastive loss (ITC) (He et al., 2020) for comparison.

| Components | | | CIRR | | FashionIQ | |
|---|---|---|---|---|---|---|
| ITC | GCA | LCR | R@5 | $R_s$@1 | R@10 | R@50 |
| *Under the zero-shot CIR setting.* | | | | | | |
| ✓ | | | 69.66 | 69.57 | 38.07 | 59.30 |
| | ✓ | | 71.64 | 69.93 | 39.04 | 59.91 |
| | ✓ | ✓ | **71.68** | **70.65** | **39.11** | **60.29** |
| *Adopt the supervised CIR setting.* | | | | | | |
| ✓ | - | | 82.92 | 79.86 | 54.53 | 74.86 |
| ✓ | - | ✓ | **83.81** | **80.87** | **54.92** | **75.55** |

Table 5: **Results with Different Datasets.** For efficiency, we train the models on 100k sampled triplets from each dataset and evaluate them under zero-shot settings.

| Dataset | Filter | CIRR | | FashionIQ | |
|---|---|---|---|---|---|
| | | R@5 | $R_s$@1 | R@10 | R@50 |
| WebVid-CoVR | - | 67.28 | 70.19 | 36.45 | 57.66 |
| LaSCo | - | 68.55 | 69.11 | 33.12 | 54.87 |
| ST18M | - | 60.00 | 57.59 | 30.60 | 51.00 |
| Independent | ✓ | 70.17 | 68.97 | 37.19 | 58.47 |
| **CIRHS (Ours)** | ✗ | 70.65 | 68.75 | 37.24 | **59.47** |
| | ✓ | **71.18** | **70.32** | **37.76** | 59.28 |

## 4.3 QUANTITATIVE RESULTS

**Comparison with supervised CIR approaches.** Table 1 presents a comparison of existing supervised CIR methods. Our method, CoAlign, achieves the best overall performance on both FashionIQ and CIRR. Specifically, SPRC focuses on sentence-level prompt optimization but lacks local understanding. CaLa (Jiang et al., 2024) aims to capture fine-grained query-target relations but suffers from suboptimal global alignment. In contrast, CoAlign adopts hybrid contextual alignment, jointly optimizing both global and local objectives in a simple yet effective manner.

## 4.4 QUALITATIVE RESULTS

Figure 4(a) presents visualizations. (1) The top two rows illustrate our model's strong multimodal reasoning, covering object composition and fine-grained semantics. (2) The bottom two rows show failure cases. Both CIRR and FashionIQ contain false negatives, e.g., the top two predictions in row 3 are actually correct. Row 4 shows an ambiguous relative caption, a common issue in FashionIQ. The model tends to prefer outputs with similar backgrounds, likely due to consistency constraints in dataset construction. Since background changes are not specified, such outputs remain acceptable.

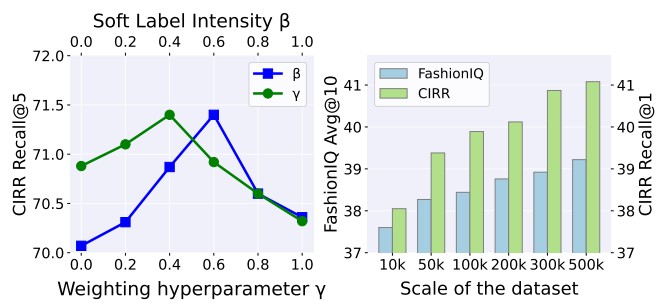

Figure 3: **Hyperparameter and data scale analysis.** Left: Sensitivity analysis of CoAlign on different hyperparameters. Right: Impact of data scale on zero-shot performance.

## 4.5 ABLATION STUDY

**CoAlign Model**. We train multiple versions of our model, as shown in Table 4. Under the zero-shot setting, we introduce the widely used image-text contrastive (ITC) loss (He et al., 2020; Li et al., 2021), which optimizes only the diagonal of the similarity matrix, to evaluate the impact of Global Contextual Alignment (GCA) in learning from broader cross-modal contexts. Results show that GCA is effective, and that combining it with Local Contextual Reasoning (LCR) further enhances performance. Similar gains are observed under supervised training. Note that in the supervised setting, where each composed query has only one target image, ITC and GCA become functionally equivalent.

**Results with Different Datasets.** To evaluate our triplet synthesis pipeline, we train CoAlign with identical settings across multiple datasets. As presented in Table 5, models trained on CIRHS outper-

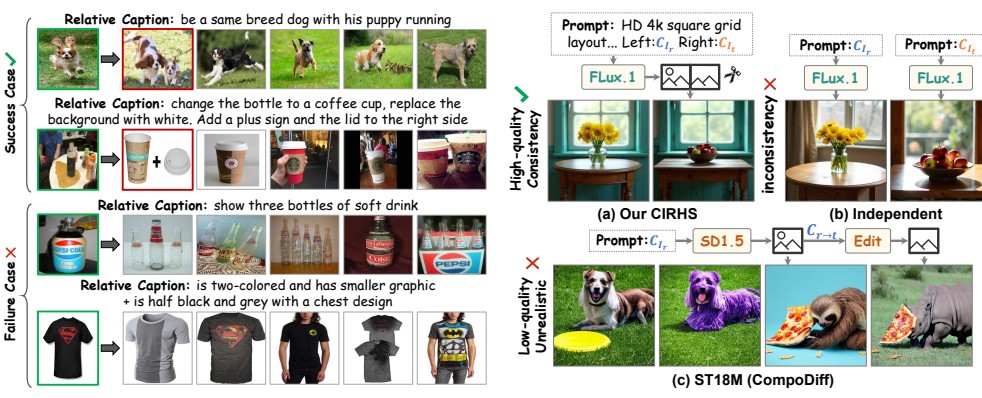

(a) Qualitative results on CIRR and FashionIQ.

(b) **Comparison of three triplet synthesis methods.** Our approach (a) outperforms (b) and (c) in both generation quality and consistency.

Figure 4: **Visualization results on FashionIQ and CIRR, as well as the visualization of three data synthesis paradigms.**

form those trained on real-world datasets like WebVid-CoVR (Ventura et al., 2024) and LaSCo (Levy et al., 2024), owing to the rich semantic diversity introduced by the LLM and the strong alignment of synthesized triplets. Compared to ST18M (Gu et al., 2024a), CIRHS mitigates issues of noise and artifacts common in image editing-based methods, yielding higher-quality reference and target images and boosting performance. We also assess an alternative strategy using independent prompts $C_{I_r}$ and $C_{I_t}$ for the T2I-GM (denoted *Independent* in Table 5), which suffers from poor consistency, making it suboptimal for CIR triplet construction. Figure 4(b) provides a visual comparison of three triplet synthesis paradigms, further supporting the above conclusion. Finally, removing low-quality samples leads to improved performance, confirming the effectiveness of our filtering strategy. Additional results using different MLLMs as filters are reported in Appendix B.

**Hyperparameter and Data Scale Analysis.** (1) The left side of Figure 3 shows the effects of $\beta$ (soft label intensity) and $\gamma$ (loss weighting hyperparameter). The performance first increases, then declines, peaking at $\beta = 0.6$ and $\gamma = 0.4$. (2) The right side shows that performance improves with more training data, saturating around 300k samples. This demonstrates that CIRHS-534k provides sufficient scale for strong CIR performance.

## 5 CONCLUSION

We propose a scalable pipeline for synthesizing high-quality CIR triplets, addressing prior limitations such as low image quality and poor semantic diversity. With this pipeline, we build a large-scale synthetic dataset, Composed Image Retrieval on High-quality Synthetic triplets (CIRHS). The pipeline employs an LLM to generate diverse, semantically rich quadruples that guide a T2I-GM to produce consistent image pairs, which are then filtered and reorganized into triplets. Furthermore, we introduce Hybrid Contextual Alignment (CoAlign), a new CIR framework that jointly optimizes global and local objectives within a broad context. Trained solely on CIRHS, CoAlign achieves strong zero-shot performance on three benchmarks. To our knowledge, it is the first CIR model trained entirely on synthetic data to reach this level of performance. Under supervised settings, CoAlign also outperforms existing methods, validating the effectiveness of our retrieval framework.

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

## A  USAGE OF LLM

In our work, the LLM is primarily used for paper refinement. Additionally, we employ it to generate high-quality textual quadruples. The details are provided in Section 3.1, and the instructions used to prompt the LLM are detailed in Section C.1.

## B  CIR BENCHMARK DETAILS

This section provides an overview of the three benchmarks, i.e., FashionIQ, CIRR, and CIRCO, involved in our study. We describe the characteristics, size, and specific tasks associated with each dataset to highlight their relevance.

**FashionIQ Wu et al. (2021)** is designed to promote conversational interfaces for online shopping, offering a more interactive approach than traditional keyword-based search systems. The dataset contains 30,134 triplets constructed from 77,684 images, categorized into three classes: Dress, Shirt, and Toptee. It also includes product descriptions and visual attribute labels. FashionIQ is split into training, validation, and test sets with a ratio of 6:2:2. Some examples of FashionIQ triplets are presented in Figure 9(a).

**CIRR Liu et al. (2021)** is developed to address the limitation of CIR benchmarks being domain-specific, like FashionIQ, by extending it to open-domain scenarios. CIRR consists of 36,554 annotated triplets, collected from a large number of visually similar images in the NLVR Suhr et al. (2019) dataset. These images are processed using ResNet-152 He et al. (2016), pre-trained on ImageNet Russakovsky et al. (2015). It is randomly divided into training, validation, and test splits (8:1:1 ratio). However, CIRR faces challenges such as ambiguous, unnecessary descriptions in the captions, and numerous false negatives (FNs), which may lead to evaluation inaccuracies. Representative examples from CIRR are illustrated in Figure 9(b).

**CIRCO Baldrati et al. (2023)** is built using images from the COCO dataset Lin et al. (2014) and is the first CIR benchmark designed specifically for zero shot CIR, where each query corresponds to multiple ground truths (average of 4.53). It consists of a validation split (220 samples) and a test split (800 samples), with no training set included. As CIRCO is designed for zero shot CIR, results on the test split are submitted to a remote server for evaluation. Its multiple ground truths provide robust metrics for mean average precision (mAP), making it a valuable benchmark for zero-shot evaluation. Figure 9(c) shows examples from CIRCO.

## C  MORE DETAILS FOR TRIPLET SYNTHESIS

A CIR triplet consists of three components: a reference image $I_r$, a relative caption $C$, and a target image $I_t$. Synthesizing such data poses two major challenges: (1) generating an image pair $(I_r, I_t)$ that shares fully overlapping elements while simultaneously modifying certain others; and (2) employing precise textual descriptions to accurately capture the relative transformations between the two images. To address these challenges, we propose an automatic pipeline, which decomposes the synthesis process into three stages. First, a large language model (LLM) is utilized to generate diversified quadruples. Second, based on the generated textual quadruples, a text-to-image generative model (T2I-GM) synthesizes the corresponding image pair, resulting in tow triplets $(I_r, C_{r \to t}, I_t)$ and $(I_t, C_{t \to r}, I_r)$. Third, we employ a multi-modal large language model (MLLM) to score and filter the generated triplets based on three aspects, discarding low-quality samples. Figure 6 illustrates the full prompts used as input to the LLM and MLLM.

### C.1  DIVERSE QUADRUPLE GENERATION

Diverse textual quadruples $(C_{I_r}, C_{r \to t}, C_{t \to r}, C_{I_t})$ are generated using an LLM. A quadruple includes the reference caption $C_{I_r}$, the target caption $C_{I_t}$, the relative caption $C_{r \to t}$ describes the modification from $C_{I_r}$ to $C_{I_t}$, while the inverse relative caption $C_{t \to r}$ captures the transformation in the opposite direction, from $C_{I_t}$ to $C_{I_r}$. To accomplish this, we adopt Qwen2.5-32B[2] Yang et al.

---

[2] Qwen/Qwen2.5-32B-Instruct

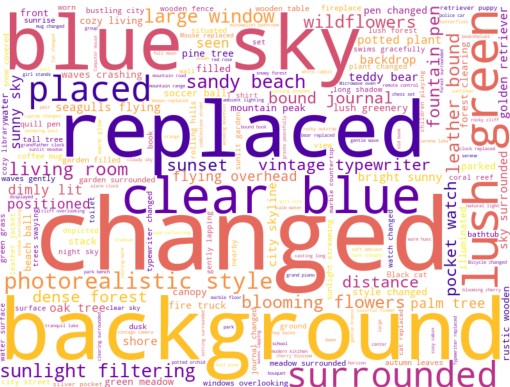

Figure 5: **A word cloud of CIRHS,** displaying all words from the relative captions according to their frequency.

(2024a) as the LLM. The relative captions cover various editing operations such as object substitution, addition, removal, quantity change, direct addressing, and viewpoint shift. The image captions, i.e., $C_{I_r}$ and $C_{I_t}$, explicitly specify the objects or scenes to be included, covering a wide range of commonly occurring items and environments in the real world. In addition, style information is embedded into the image captions, enabling the synthesis of data across multiple domains and thereby enhancing the robustness of CIRHS. As shown in Figure 6, each prompt ensures consistent format and prefix.

For example, in cases involving object substation, the prompts clarify that $C_{I_r}$ and $C_{I_t}$ should describe different objects while keeping all other elements unchanged. Moreover, $C_{r \to t}$ and $C_{t \to r}$ must only reflect the differences between $C_{I_r}$ and $C_{I_t}$. To further enhance output quality and diversity, each prompt includes three high-quality annotated examples, randomly sampled from a curated set of 100 high-quality triplets generated by GPT-4o OpenAI et al. (2024). These examples not only stabilize the LLM's output but also reduce redundancy caused by similar input prompts. Each prompt contains an object randomly drawn from a set of 200 common categories, also generated by GPT-4o, and a style selected from 20 predefined options, with an emphasis on realism while preserving diversity. Finally, the LLM generates structured outputs that serve as inputs for the synthesis of consistent image pair.

### C.2 CONSISTENT IMAGE PAIR SYNTHESIS

Given a textual quadruple $(C_{I_r}, C_{r \to t}, C_{t \to r}, C_{I_t})$, we synthesize a pair of images $(I_r, I_t)$ by transforming $C_{I_r}$ and $C_{I_t}$ into corresponding images. Due to the stochastic nature of diffusion models, the key challenge lies in maintaining consistency across shared elements in both images. Fortunately, diffusion models inherently support the generation of identical objects within a single image, an ability commonly referred to as the in-context capability Hui et al. (2025). We exploit this property by synthesizing a single image containing two side-by-side sub-images, ensuring high visual consistency of shared elements across $I_r$ and $I_t$.

To implement this, $C_{I_r}$ and $C_{I_t}$ are embedded into a side-by-side prompt template and fed into the T2I-GM, for which we adopt FLUX.1-dev[3] Labs (2024), producing a $512 \times 1056$ resolution image. This configuration allows us to split the image into two $512 \times 512$ sub-images, forming the desired pair $(I_r, I_t)$. The added padding helps mitigate artifacts from cropping and ensures higher visual fidelity. As illustrated in Figure 6, this approach offers significant advantages over independent generation, as it better preserves shared visual elements while allowing for controlled differences. Each quadruple is used to generate 10 $(I_r, I_t)$ pairs, which are combined with relative captions in both directions, resulting in 20 triplets per quadruple. This design makes our data synthesis pipeline highly efficient. Among the 20 generated triplets, the two sets of 10 triplets in the same direction are theoretically equivalent. However, some visual differences may still exist across generated image

---

[3]black-forest-labs/FLUX.1-dev

Figure 6: **Pipeline of high-quality triplet construction with detailed instruction design.** The illustrated example demonstrates the generation of a triplet involving an object substitution. The construction for other types of editing operation follow a similar design paradigm.

pairs. To address this, we introduce the Triplet IDentity (TID), where triplets sharing the same relative caption are assigned the same TID, allowing label smoothing Müller et al. (2019) to be applied among triplets sharing the same TID.

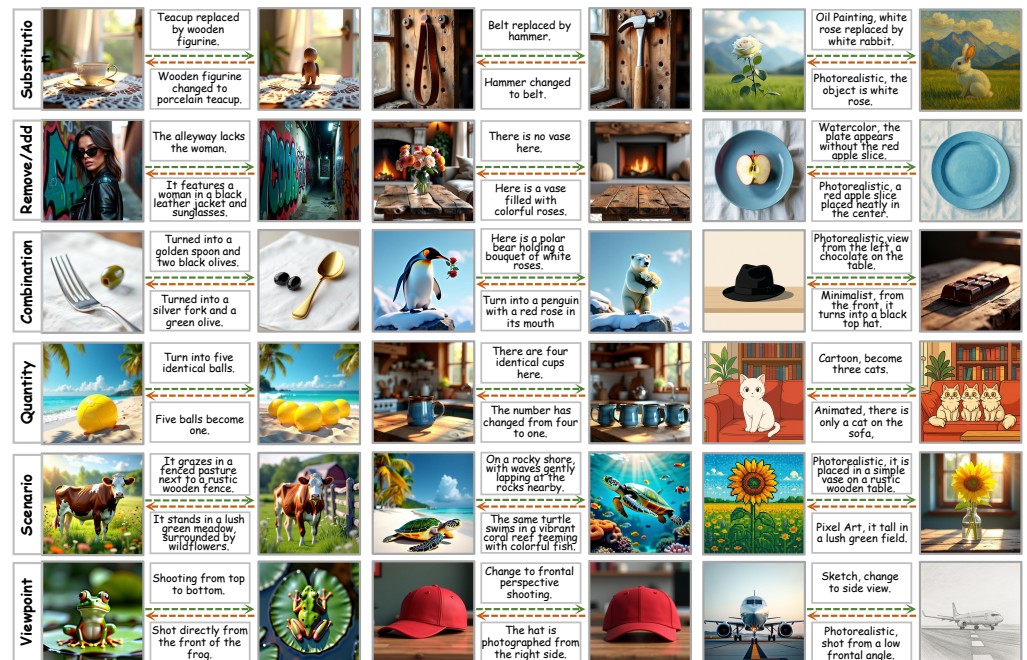

Figure 7: **Additional examples from the CIRHS dataset.** Each row presents a typical type of editing operation, and each example serves as two CIR triplets.

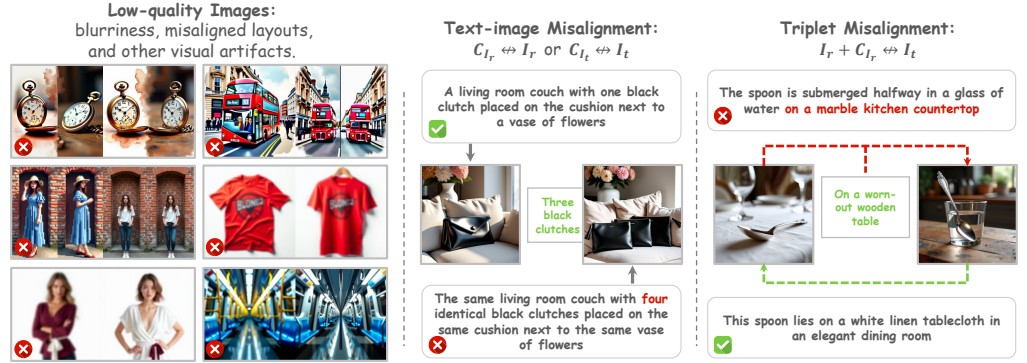

Figure 8: **Some low-quality samples.** The figure presents examples from three dimensions. Left: Samples with low image quality scores, mainly exhibiting issues such as blurriness, incorrect layout, and other visual artifacts. Middle: Image–text misalignment, where the generated images do not correspond well to the input captions used for synthesis. Right: Triplet inconsistency, where the relationship between the reference and target images fails to match the relative caption.

## C.3    DATA FILTERING

To ensure the overall quality, we employ an MLLM, namely Qwen2.5-VL-32B[4] Bai et al. (2025), with carefully designed evaluation prompts to score and filter the synthesized triplets. The evaluation considers three dimensions: (1) Image quality, which assesses clarity, noise, and artifacts in both $I_r$ and $I_t$; (2) Image-text fidelity, which measures how well the images match the textual descriptions, i.e., $I_r \leftrightarrow C_{I_r}$, $I_t \leftrightarrow C_{I_t}$; and (3) Triplet alignment, which evaluates whether $I_r$ and $C_{r \to t}$ together can accurately infer $I_t$. The MLLM assigns scores from 1 to 10 for each dimension and aggregates them into an average score. Based on this score, we filter the generated triplets and retain only high-quality samples in the CIRHS dataset. Some low-quality samples are illustrated in Figure 8. In

---

[4]Qwen/Qwen2.5-VL-32B-Instruct

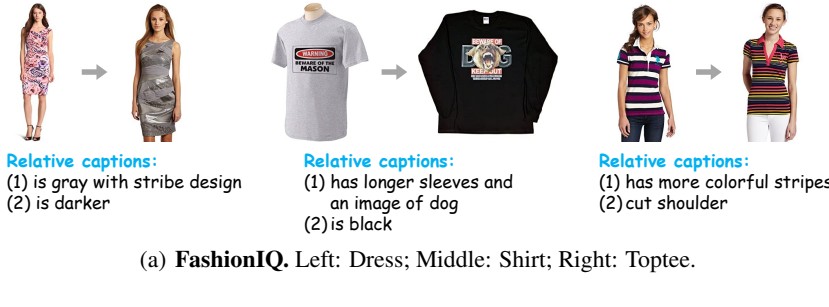

Relative captions:
(1) is gray with stribe design
(2) is darker

Relative captions:
(1) has longer sleeves and
an image of dog
(2) is black

Relative captions:
(1) has more colorful stripes
(2) cut shoulder

(a) **FashionIQ.** Left: Dress; Middle: Shirt; Right: Toptee.

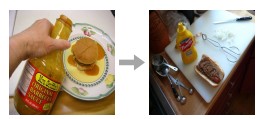
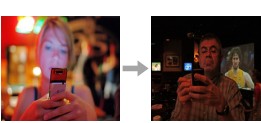
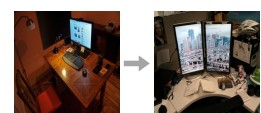

Relative caption:
The background water is
missing and the focus is on
the man with the three dogs

Relative caption:
has one cheetah looking to
the camera and yellow
flowers the left side

Relative caption:
Stack up the white cups

(b) **CIRR,** which is the first open-domain CIR dataset.

Relative caption:
is resting on a table and
the photo has a hot dog
instead of a burger

Relative caption:
has a screen in the
background and is darker

Relative caption:
has two of them and they
are vertical

(c) **CIRCO,** which is also a real-world dataset.

Figure 9: **Representative examples from the three CIR benchmarks.** For each example, the reference image is shown on the left, the target image on the right, and the corresponding relative caption is displayed below.

Table 6: **Results using different MLLMs as filtering modules.** For efficiency, we train the models on 100k sampled triplets from CIRHS and the evaluation is based on zero-shot settings.

| Filtering Model | CIRR | | | FashionIQ | |
|---|---|---|---|---|---|
| | R@1 | R@5 | R$_s$@1 | R@10 | R@50 |
| Base (w/o filter) | 38.46 | 70.65 | 68.75 | 37.24 | 59.47 |
| Ovis2-16B | 40.19 | 71.23 | 69.66 | 37.57 | 59.76 |
| InternVL3-14B | 40.23 | 71.02 | 70.48 | 37.64 | 59.19 |
| Qwen2.5-VL-7B | 39.49 | 70.98 | 69.27 | 37.11 | 58.97 |
| Qwen2.5-VL-32B | 41.17 | 71.18 | 70.32 | 37.76 | 59.28 |

addition, we evaluate the performance of different MLLMs as filtering modules, including Ovis2 Lu et al. (2024), InternVL Chen et al. (2024b), and Qwen2.5-VL Bai et al. (2025). As shown in Table X, all models lead to performance improvements when used for filtering, though the differences among them are relatively minor. Considering both stability and scalability, we adopt Qwen2.5-VL-32B, the most widely used model in the open-source community, as our final filtering module.

## C.4 SAMPLE ILLUSTRATIONS FROM THE CIRHS DATASET

Figure 7 shows representative samples in the CIRHS dataset, demonstrating its high quality and semantic diversity. Notably, the rightmost column showcases examples involving cross-domain image pairs, a unique feature of CIRHS not covered by existing CIR datasets. In addition, as presented in Figure 5, we visualize the relative captions in the CIRHS dataset as a word cloud based on word frequency, illustrating a wide range of editing operations as well as real-world objects and scenes.

