# OpenReview forum: "Automatic Synthesis of High-quality Triplet Data for Composed Image Retrieval"
_ICLR.cc/2026/Conference — ICLR 2026 Conference Withdrawn Submission_

### Official Review · Reviewer_gPsB · 2025-10-17

**Soundness:** 3
**Presentation:** 4
**Contribution:** 3
**Rating:** 4
**Confidence:** 3

**Summary:**

This paper proposes a scalable pipeline for automatic triplet generation for the composed image retrieval task. A novel synthetic dataset, CIRHS, is generated with the pipeline. In addition, it proposed a framework with a novel hybrid contextual alignment module. Using both components, it achieved good zero-shot performance on a common benchmark. Its performance can be further improved with supervised training.

**Strengths:**

- This work proposed a CoAlign framework that considers both global alignment and local reasoning for the CIR task. The proposed approach is simple yet effective.
- This paper demonstrated that with high quality synthetic dataset, it is possible to achieve strong CIR performance under a zero-shot setting. The work conducts a comprehensive evaluation and justifies the effectiveness of the proposed synthetic dataset generation pipeline and CIR framework.

**Weaknesses:**

- The work claims this work is the first to demonstrate the feasibility of training CIR models on a fully synthetic dataset; this might be overclaimed. In CompoDiff, Gu et al. 2024a, a synthetic dataset (SynthTriplets18M) is created and used for training the CIR model. In "Improving Context Understanding in Multimodal Large Language Models via Multimodal Composition Learning", ICML 2024, a text-based synthetic data generation approach is applied for existing image-caption pairs for training multimodal LLMs.
- The high-quality synthetic triplets generation leveraged strong LLMs and image generation models. A pipeline is created with various considerations; however, the novelty of generating a high-quality dataset is limited, as a similar idea was proposed in CompoDiff, Gu et al. 2024a. It would be good to discuss whether CIRHS benefited from better LLM, diffusion model, or the generation pipeline.
- The CoAlign framework's technical novelty is modest. The novelty lies more in the integration than in introducing fundamentally new techniques. I acknowledge that CoAlign shows effectiveness in the common benchmark.

**Questions:**

- I would like the author to discuss why ST18M is unable to generate high-quality images. Is it due to the underlying image generation/editing models, or other factors? Compared to ST18M, has the generated CIRHS benefited from better LLM, diffusion model, or the generation pipeline?
- In Table 2, what is the number of images of ST18M? In line 323, it is stated that it suffers from poor generation quality due to unrealistic outputs and visual artifacts. Despite that, the author should still provide the number of images and image samples for comparison.
- Missing bibliographic details and duplicate entries were identified during the review process.

---

### Official Review · Reviewer_Dh1q · 2025-11-01

**Soundness:** 3
**Presentation:** 3
**Contribution:** 2
**Rating:** 4
**Confidence:** 3

**Summary:**

The paper addresses the issue of low-quality synthetic triplets in composed image retrieval (CIR) by proposing a fully synthetic dataset called CIRHS (Composed Image Retrieval on High-quality Synthetic Triplets), constructed through a three-stage pipeline: (1) diversified synthesis, (2) consistent image-pair synthesis and quadruple generation, and (3) data filtering. Based on this dataset, the authors further propose CoAlign (Hybrid Contextual Alignment), a CIR framework designed to achieve both global alignment and local reasoning. Extensive experiments demonstrate that the proposed dataset and method yield significant improvements in both zero-shot and supervised settings.

**Strengths:**

1. The paper is well-written and well-structured, making it easy to follow and understand the main contributions.
2. The proposed generation pipeline is well-motivated, and the method itself is simple yet effective.
3. The experiments are carefully designed to demonstrate the effectiveness of the approach, and the proposed method shows strong and consistent performance across diverse settings.

**Weaknesses:**

1. I believe the main decision point of this paper lies in the contribution of the newly generated dataset.
- As shown in Table 5, there already exist numerous prior works that synthetically construct CIR triplets using LLMs, MLLMs, T2I models, or collection-based approaches. The paper should better highlight the differences between the proposed dataset and these existing methods, perhaps with a comparative summary table. While the results demonstrate the effectiveness of the proposed training dataset (showing strong performance despite a smaller scale), it remains unclear whether the improvement genuinely stems from the proposed pipeline or simply from using more advanced and recent LLMs/MLLMs. Although Table 5 includes partial analyses, a more concrete and detailed comparison would strengthen the paper’s contribution. Considering that many previous works share similar motivations and methods, additional discussion and evidence are necessary to clarify what truly differentiates this dataset.
- Moreover, comparisons with relevant recent works such as CoVR-2 (Ventura et al., Automatic Data Construction for Composed Video Retrieval) and MagicLens (Zhang et al., Self-Supervised Image Retrieval with Open-Ended Instructions), as well as Compodiff, would provide a clearer positioning of this work within the existing literature.
- It would be valuable to include a scaling analysis of the proposed dataset—examining performance trends as the dataset size increases (e.g., from smaller subsets to sizes larger than the current 500K).

2. The proposed method appears effective, but its improvement and novelty should be more clearly explained.
- In Table 4, the results for LCR are not fully convincing. The standalone effect of LCR seems minimal when combined with GCA. It would be helpful to isolate its individual contribution through ablation or discussion.
- Further clarification is also needed regarding GCA, particularly the computation of β in lines 268–269. Does this refer to a form of label smoothing? If the contribution is indeed related to a simple label-smoothing variant, it may not be highly novel; however, since it yields non-negligible performance gains, a more in-depth analysis explaining why it works would be valuable to justify its contribution.

**Questions:**

Wrote above

---

### Official Review · Reviewer_6GiR · 2025-11-04

**Soundness:** 3
**Presentation:** 3
**Contribution:** 2
**Rating:** 2
**Confidence:** 4

**Summary:**

This paper proposes a synthetic CIRHS dataset and a hybrid contextual contextual alignment (CoAlign) method for the task of composed image retrieval, which leverages LLM to generate diverse prompts, and accomplishes global alignment and local reasoning within a broad context.

**Strengths:**

1. The paper is well written and easy to follow.
2. Figure illustrations clearly demonstrate the motivation of the paper and the data generation pipeline.

**Weaknesses:**

1. The paper introduces a new CIRHS dataset, but no results concerning the CIRHS dataset are provided, making the significance of the dataset and the contribution of the paper unclear.
2. The core idea of global and local alignment in the CoAlign method is not new, more theoretical insights should be provided to strengthen the novelty of the proposed method.

**Questions:**

1. More comparison results concerning the proposed CIRHS dataset should be provided, including CIR performance and cross-domain performance.
2. More theoretical insights should be provided to strengthen the novelty of the proposed method.
3. The organization of the paper should be inproved, such as aligning the tables for better presentation.

---

### Official Review · Reviewer_Dvxq · 2025-11-06

**Soundness:** 3
**Presentation:** 2
**Contribution:** 2
**Rating:** 2
**Confidence:** 5

**Summary:**

This paper details a scalable pipeline that automatically synthesizes high-quality triplet datasets for Composed Image Retrieval (CIR). The process leverages a series of advanced models, starting with an LLM to generate diverse textual quadruplets. A Text-to-Image model then produces visually consistent query-target pairs using an effective technique of generating and splitting a single composite image. A Multimodal LLM subsequently filters these generated triplets to ensure data quality. Furthermore, the paper proposes a "Hybrid Contextual Alignment" method, specifically tailored for this synthetic data, which significantly boosts training performance. This comprehensive approach allows the model to achieve superior results, outperforming existing works even when trained on a comparatively smaller dataset.

**Strengths:**

- High scalability and engineering efficiency
This paper constructs a data generation pipeline using state-of-the-art open-source models (LLM, T2I generative model, MLLM). This implies that the scale of the dataset can be continuously expanded, provided that sufficient GPU resources are available. Furthermore, the data generation process is more engineering-friendly and simpler compared to previous methods like 'compodiff', making it highly suitable for adoption in other research efforts aiming to build synthetic datasets.

- Simple yet effective consistency maintenance of two generated images
High visual consistency between query and target images is important for the quality of a CIR dataset. The paper addresses this challenge with a simple yet effective approach: generating a single composite image and then splitting it. This way is expected to make significant contributions to similar data generation tasks where consistency is paramount.

- The paper clearly demonstrates through quantitative experiments that its proposed dataset and learning methodology achieve superior performance compared to models trained on other existing datasets and data generation methods.

**Weaknesses:**

The contributions emphasized in this paper are summarized as follows:
- Proposed a scalable synthesis pipeline to address prior data limitations like **low image quality, unrealistic appearances, and the lack of diversity in captions**, resulting in the large-scale, high-quality CIRHS dataset (534k triplets).
- Introduced a novel CIR framework, **Hybrid Contextual Alignment (CoAlign)**, which enhances learned representations by combining global alignment and local reasoning.
- Experiments demonstrate superior performance and assert that this work is the first **to verify the feasibility of training CIR models entirely on synthetic data**.

However, I have the following concerns regarding these contributions (see the above text in bold):

I believe the contribution regarding the high quality of the generated images is insufficient because the authors used a recent high-quality text-to-image generative model. CompoDiff [1], which the paper compares against, used an older model (Stable Diffusion [2,3]). If CompoDiff had used Flux [4] in the same manner, I believe the issue of low image quality could have been resolved by CompoDiff as well.

The claim regarding the diversity of the proposed dataset is not substantiated. The authors criticize other datasets for lacking diversity and claim their proposed data has higher diversity, but this is not supported by any statistical information or experiments. Such a claim is not acceptable without empirical evidence.

Furthermore, the motivation for proposing a loss tailored to this data is not convincing. For example, the paper lacks experimental results for a scenario where batch sampling is performed by extracting from different TIDs (Triplet Identities) to ensure no duplicate TIDs exist within a batch. Without these results, the argument that the proposed loss must be adopted for batches containing duplicate TIDs is not persuasive.

Moreover, I strongly disagree with the claim that this is the first work to verify "the feasibility of training CIR models entirely on synthetic data." This claim is entirely unsubstantiated.

The paper is missing references to recent relevant dataset proposals such as MegicLens [5] and MegaPairs [6]. Furthermore, this paper still addresses a problem definition that these preceding works aimed to overcome (i.e., breaking the strict consistency between the query image and target image to reflect higher-level queries). For example, the proposed method likely cannot answer a query like "What restaurants are in this area?" given a query photo of the Empire State Building. Therefore, the timeliness and novelty of this paper are questionable.

[1] CompoDiff: https://arxiv.org/abs/2303.11916

[2] SD1.5: https://huggingface.co/stable-diffusion-v1-5/stable-diffusion-v1-5

[3] SD2: https://huggingface.co/stabilityai/stable-diffusion-2

[4] Flux models: https://huggingface.co/black-forest-labs/models

[5] MagicLens: https://arxiv.org/abs/2403.19651

[6] MegaPairs: https://arxiv.org/abs/2412.14475

**Questions:**

Please provide your response to the Weaknesses section. I would appreciate it if you could show results that address my concerns.

The performance will vary depending on which backbone was used. Are there any results of training on CLIP instead of BLIP?

---

### Note · Authors · 2025-11-14

I have read and agree with the venue's withdrawal policy on behalf of myself and my co-authors.